# “Everyone Has the Right to Drink Beer”: A Stakeholder Analysis of Challenges to Youth Alcohol Harm-Reduction Policies in Lebanon

**DOI:** 10.3390/ijerph16162874

**Published:** 2019-08-12

**Authors:** Rima T Nakkash, Lilian A Ghandour, Nasser Yassin, Sirine Anouti, Ali Chalak, Sara Chehab, Aida El-Aily, Rima A Afifi

**Affiliations:** 1Department of Health Promotion and Community Health, Faculty of Health Sciences, American University of Beirut, Beirut 1107-2020, Lebanon; 2Department of Epidemiology and Population Health, Faculty of Health Sciences, American University of Beirut, Beirut 1107-2020, Lebanon; 3Department of Health Management and Policy, Faculty of Health Sciences, American University of Beirut, Beirut 1107-2020, Lebanon; 4Department of Agriculture, Faculty of Agricultural and Food Sciences, American University of Beirut, Beirut 1107-2020, Lebanon; 5Department of Community and Behavioral Health, College of Public Health, University of Iowa, IA 52242, USA

**Keywords:** alcohol, youth, alcohol advertising, drinking and driving, alcohol control policy, alcohol industry, Lebanon, Arab

## Abstract

Background: Alcohol use is a major risk factor in premature death and disability, especially among youth. Evidence-based policies to prevent and control the detrimental effect of alcohol use have been recommended. In countries with weak alcohol control policies—such as Lebanon, stakeholder analysis provides critical information to influence policy interventions. This paper assesses the views of stakeholders regarding a national alcohol harm reduction policy for youth. Methods: We interviewed a total of 22 key stakeholders over a period of 8 months in 2015. Stakeholders were selected purposively, to include representatives of governmental and non-governmental organizations and industry that could answer questions related to core intervention areas: affordability, availability, regulation of marketing, and drinking and driving. We analyzed interview transcripts using thematic analysis. Results: Three themes emerged: Inadequacy of current alcohol control policies; weak governance and disregard for rule of law as a determinant of the status quo; and diverting of responsibility towards ‘other’ stakeholders. In addition, industry representatives argued against evidence-based policies using time-worn strategies identified globally. Conclusions: Our findings indicate that alcohol harm reduction policies are far from becoming a policy priority in Lebanon. There is a clear need to shift the narrative from victim blaming to structural conditions.

## 1. Introduction

Alcohol use is a leading risk factor for premature death and disability [1] given its link to more than 60 non-communicable diseases and injuries [2]. Alcohol use attributable Disability-Adjusted Life Years (DALYs) increased by more than 25% from 1990 till 2016, accounting for 4.2% and 5.2% of the total DALYs and death, respectively [3]. In 2016, the highest proportions (13.5%) of all alcohol-attributable deaths occurred among young people aged 20–39 years [4]. Youth alcohol drinking is a particular major concern given the acute effects of binge drinking [5], and associations with unintentional injuries, including drunk-driving crashes, suicide, and violence [6,7]. Early onset of alcohol consumption has also been consistently and strongly linked to risky youth behaviors, including risky sexual practices [8,9,10], and gambling [11]. Early onset of alcohol use has also been associated with an increased risk of developing health problems such as non-communicable diseases [12], and mental and behavioral disorders, including alcohol and other substance use disorders in adulthood [13,14,15,16,17,18,19].

Despite gradual declines in youth alcohol consumption over the past two decades in many developed countries, alcohol use among high-school and college students remains high [20,21]. Lebanon is no exception. While international guidelines recommend that adolescents less than 15 years of age not drink alcohol, and that those aged 15–17 delay alcohol initiation as long as possible [22,23], local epidemiological data in Lebanon revealed that between 2005 and 2011, past 30-day alcohol use increased by 40% among 7th–9th graders and lifetime drunkenness by 50% [24]. The latest data from 2017, further reveal that 83% of the middle school students (13–15 years) who have ever drank in their lifetime admitted to having had their first alcoholic drink before age 14 [25]. In addition, approximately 1 in 5 middle school students (18.9%) were past 30-day drinkers, and 13.4% of the lifetime drinkers admitted that they had so much alcohol that they were really drunk one or more times during their life. Alcohol drinking in minors in Lebanon is seemingly more regular than occasional. In a sample of high school students (aged 16.78 ± 0.06 years) enrolled in private, public, as well as vocational schools in the greater Beirut area, 40% of past-year drinkers reported drinking alcohol once or twice per week or more [26].

More generally, and as per the WHO Global Alcohol Status Report, the total per capita consumption of alcohol (in liters) of Lebanese adults (15+ years) was 2.46 in 2010, and this consumption amount is exceeded only by that of Bahrain and Sudan in the Arab region and is equal to that of Qatar [27]. The patterns of drinking score is an indicator that measures ‘riskiness’ of those patterns, with a higher score indicating more risk (range: 1–5). Data for Arab countries is limited, with Lebanon and 2 other countries (Djibouti and Sudan) scoring a 3, and 2 other countries scoring a 2 (Tunisia and Algeria). In the Arab world, alcohol consumption is evidently not considered a priority for research or policy-making across most, if not all, Arab nations. Research and scholarship on alcohol consumption and harm (indicated by quantity and coverage) is strikingly low with only 69 articles published across 20 years and 22 countries [27]. The published data is hampered by the biases inherent in the samples studied as well as lack of comparability of studies within and across countries.

The World Health Organization [4] recommends a series of evidence-based policies to prevent and control the detrimental effect of alcohol use [28]. These include four core policies: affordability (e.g., pricing and taxation of alcohol beverages); availability (e.g., minimum legal drinking age); regulation of marketing (e.g., alcohol advertising and marketing including sponsorship of events); and drinking and driving [28]. A recent in-depth analytical review of Lebanon’s alcohol control policy found that regulations governing alcohol are largely non-existent or existent but very outdated and loosely enforced [27]. With specific reference to some of the WHO policies, Lebanon does not regulate the legal alcohol purchase age, although decrees exist that prohibit youth who are less than 18 years of age from entering bars and clubs of all kinds during day and night time, and impose a penalty fine (4–14 USD) on persons who causes minors under 18 years of age to get drunk by offering them spirit drinks. Policies for taxation and pricing do not include stipulations pertaining to imposing a ‘‘minimum price’’ and ‘‘price-related measures’’/price index specific to the different alcoholic beverages. Recently (2012), a blood alcohol content limit was imposed regulating drinking and driving [27].

Evidence suggests that a variety of factors influence the passage and implementation of policies [29,30,31]. One key assessment tool includes a stakeholder analysis; a method of research that aims to generate knowledge about behaviors, interests, and perceptions of key actors vis-a-vis a specific issue [32]. A stakeholder analysis weighs the ‘power’ of a stakeholder to their specific position (positive or negative) on a policy to understand the likelihood that the policy in question will be passed, and/or implemented and enforced. ‘Stakeholders’ is a broad term that identifies all the ‘actors’ in a particular policy context and for a specific policy, and in the case of alcohol control policies, stakeholders can include: representatives of various governmental organizations (health, economy, tourism, and internal security), non-governmental organizations (NGOs) that work to prevent harms from alcohol, the hospitality sector, and industry [32]. Published evidence has particularly highlighted the influence of the alcohol industry on policy making [33,34,35,36] and a recent systematic review concluded that the industry has consistently framed policy debates to sustain their commercial interests [37].

In this paper, we aim to assess the views of key stakeholders from the government, civil society organizations and the alcohol industry regarding a national alcohol harm reduction policy aimed at reducing the consequences of youth harmful alcohol use in Lebanon. Though the WHO policies are not specifically aimed at youth harm reduction but rather general harm reduction, we approached stakeholders from that perspective presuming it would be a more neutral position (all would agree to the need to ‘protect’ young people), and to minimize concerns—in an increasingly polarized religious context—that we might be advocating for prohibition.

## 2. Materials and Methods

### 2.1. Context

The republic of Lebanon is a small country (10,400 km^2^) on the Mediterranean sea; Beirut is its capital city. Although no official census has been conducted since 1932, the population of Lebanon is estimated at 4.4 million [38]. Lebanon was a French mandate country between 1920 and 1943 when it gained its independence. Yet, much of the influence of this French mandate is still felt, in laws that were established during that period and remain only marginally changed, and in the system of confessionalism/sectarianism that pervades the country. Lebanon is one of the most religiously diverse countries of the Arab region, with 18 official religious sects recognized by the government; most following either Islam (which prohibits alcohol use) or Christianity. A sectarian system governs all aspects of Lebanese formal and informal interactions [39]. Indeed the sectarian system is the social capital used by citizens to obtain resources and rights in education, inheritance, employment, and other mechanisms [40]. In addition, Lebanon has been the hotbed of regional and national conflicts. Political events were frequent between 1975 and 2011. A ‘chronology of key events in the history of Lebanon’ provided by the BBC clearly demonstrates the continuous uncertainty and crises [41]. The perception and realities described by the stakeholders we interviewed may be more fully understood with this context as background.

### 2.2. Stakeholder Selection

A total of 22 semi-structured interviews with key stakeholders were carried out over a period of 8 months in 2015. Stakeholders were selected purposively, with specific intent to include representatives of governmental agencies, NGOs, as well as industry that could answer questions related to four core intervention areas identified by WHO: affordability (e.g., pricing and taxation of alcohol beverages); availability (e.g., minimum legal drinking age); regulation of marketing (e.g., alcohol advertising and marketing including sponsorship of events); and drinking and driving [28]. The research team compiled a list of potential stakeholders with expertise in one of the four core intervention areas of WHO, and that had publicly available contact information.

### 2.3. Interview Process

The Principal Investigator (Lilian Ghandour) sent a letter to each of the selected stakeholders requesting an appointment with him/her. The letter briefly explained the general objectives of the meeting, and was followed up within a week by a phone call to set an appointment. Following the phone call, those who were unavailable or whom we were not successful in reaching were replaced by another option from the list. For the stakeholders who agreed to a meeting, a time and place was scheduled at their convenience. After answering all the questions, stakeholders were asked to suggest other important stakeholders to interview. These were contacted after the research team confirmed their relevance to the research. Of the 34 stakeholders that were on the original sampling frame, 22 agreed to be interviewed. Written consent was obtained from all interviewed stakeholders (Table 1). The interviews were conducted in Arabic by one member of the research team (Lilian Ghandour, Rima Nakkash, Nasser Yassin, Ali Chalak, Rima Afifi, and Aida El-Aily-), and notes were recorded by the research assistant. Nineteen stakeholders gave permission to record the interview and these were subsequently transcribed in their original language (Arabic). For the remaining interviews, extensive notes were taken by the research assistant.

### 2.4. Interview Guide

A structured interview guide was developed to ensure coverage of relevant issues in a systematic manner and allow for probing during the interview process. Questions addressed the four WHO core policy domains (advertising/marketing, availability, affordability, and drink-driving). Questions were also asked about challenges to formulating and implementing strong alcohol control policies. Each interview began with the following question: “What is your opinion of the current status of national alcohol harm reduction policies, especially with regard to youth?” The interview guide was then tailored to the expertise of stakeholders. For example, a stakeholder from the Ministry of Finance would be asked questions specifically related to fiscal policies, while one from the Ministry of Tourism would be asked questions related to availability and accessibility issues in alcohol serving establishments.

### 2.5. Analysis

We conducted thematic analysis, which is an approach in qualitative research that entails “identifying, analyzing, organizing, and describing” data collected [42]. This was achieved via a series of steps. First, all research team members individually read a common set of interviews to familiarize themselves with the data. Following that, in a series of meetings, they discussed and agreed on initial codes. The research assistant (SA) then completed coding the transcripts and conducted the thematic analysis accordingly. As a final step, the research team met again to review the themes and discuss the findings, which led to the generation of the themes presented in this manuscript. Thematic analysis was conducted in Arabic. Quotes selected for this paper were then translated into English by the research team.

### 2.6. Ethical Considerations

All subjects gave their informed consent for inclusion before they participated in the study. The American University of Beirut’s Institutional Review Board approved the protocol for this research (AUB IRB ID code: FHS.LG.13).

## 3. Results

We report the views of interviewed stakeholders in relation to existing alcohol harm reduction policies in Lebanon. Three themes emerged from our analysis of the data: (1) Inadequacy of current alcohol control policies; (2) weak governance and disregard for rule of law as a main determinant of the status quo; and (3) diverting of responsibility towards ‘other’ stakeholders. We use specific quotations from interviewees to further elaborate on the themes; but only refer to the interviewee by their organizational membership: Governmental organization (GO), Non-governmental organization/Civil Society (NGO) or Alcohol Industry (AI). We present the views of the stakeholders representing the alcohol industry as a separate theme but link their responses to the three overall themes where applicable.

### 3.1. Inadequacy of Current Alcohol Control Policies

According to interviewed stakeholders, existing alcohol policy laws in Lebanon were “old and ineffective “(GO2a) as a strategy for the prevention and control of youth alcohol use. Interviewees saw the purpose of the current alcohol policies as “A way of making money [in reference to alcohol trade]” (NGO2). Alcohol was perceived to be a largely “deregulated” commodity (NGO2); regulations were weak or nonexistent in terms of access to those who were underage, availability, affordability, drink driving, and restrictions to marketing/advertising.

With regards to *alcohol availability* and access to the underage drinker, one interviewee noted:
“[People have] all the freedom to buy alcohol from small shops whenever you want, both night and day, and to drink alcohol wherever you want, and whichever age you want”.(NGO2)

Another noted the lack of appropriate urban planning as problematic since, “commercial, touristic, and residential areas are not separated; there is an overlap between residential and non-residential areas,” which they felt made alcohol more accessible (GO3). This stakeholder pointed to an existing law (issued in 2000), that allows for on-premise alcohol selling outlets to be at 20 m proximity from schools. S/he noted that this was too close a distance to schools, and therefore was an inadequate measure to protect youth from access to alcohol.

Stakeholders further noted that in outlets like big supermarkets and minimarkets, there is virtually no control over underage purchasing of alcohol and local alcohol take-away shops are available and ubiquitous. In fact, youth accessibility was alarming as reflected by one stakeholder:
“I can send my 11-year-old daughter to the supermarket to buy a bottle of whisky, and they would give it to her.”.(NGO1)

*Alcohol affordability* was also unregulated, as interviewees noted the:
“Difficulty [in] controlling alcohol sale in small convenience stores where any person can enter to buy alcoholic products at a very cheap price as low as Lebanese Pounds/Liras 1000 (USD 0.66) which is very affordable to underage youth and promotes pre-drinking/pre-loading among youth. The problem is that the managers of the convenience stores would argue that they are licensed to sell alcohol [licensed by Ministry of finance and regulated by the Ministry of Economy and Trade] and nothing can stop them”.(GO3)

Related to *drinking and driving*, stakeholders noted that the more recent 2012 transportation law, which prohibits driving under the influence, is inadequately enforced. In addition, a local NGO stakeholder noted that, “alcohol misuse [while drinking and driving] was regarded until recently as an act of neglect [by the driver], and not considered as a criminal act in itself” (NGO2).

In terms of *alcohol advertising*, stakeholders noted that alcohol TV commercials heavily dominated local airtime, supported by large budgets and featuring youth partaking in alcohol drinking as part of daily life:
“Advertising is a con… and I call it false advertising. Today the advertising company responsible for alcohol advertisements holds a large account and focuses on teenagers who are 16 and 17 years old because they are a long-term investment. They convey motorcycles, pretty girls, nice cars.”.(NGO3)

The same stakeholder added that alcohol companies sponsor many Lebanese sports leagues with TV advertisements where they “pay 0.5 million dollars for one sponsoring activity and they have a lobby group of sports journalists on their side” (NGO3). A senior governmental official concurred that:
“The existing decree for regulating advertising of such harmful products [3] is not effective”.(GO5)

In fact, there is no specific law in Lebanon that specifically regulates alcohol advertising.

### 3.2. Weak Governance and Disregard for Rule of Law as Main Determinants of the Status Quo

The above status quo was widely attributed to the weak governance framework and disregard for the rule of law in Lebanon, specifically in relation to alcohol policy control, but more generally on all other policy-related aspects. The governmental ‘system’ has “a lot of flaws” and is beset by “all kinds of problems” (GO7). Stakeholders pointed to the poor responsiveness of key governmental agencies when it comes to the implementation of necessary alcohol control policies, “some laws and decrees exist with lack of monitoring and no consequences to violations.” (GO4).

More generally, stakeholders noted that weak enforcement of most laws in Lebanon resulted in a “culture encourage(ing) breaking (of) laws” (GO1) whereas another added:
“We always go back to the same issue, we [the country] have gotten used to this chaos [in reference to weak rule of law and corruption]”.(GO2a)

One government official commented on an incident when in 2012, the minister of interior deliberately encouraged citizens to flout the tobacco control policy. He stated the need for:
“A system that imposes penalties [on those who break the law], as it is not acceptable for a Minister who is supposed to implement a law to ban smoking in public places, to announce a hiatus allowing smoking on New Year’s Eve”.(GO7)

In addition to the role of the government, citizens also were seen to have a role in promoting enforcement:
“You need someone to complain, but if you are neglecting violations, and then citizens don’t complain, nobody complains”.(GO7)

Stakeholders also pointed to inadequacies of the judicial system, which “also needs awareness, and part of the evolution of the law in any country is how aware their judges are about society’s problems” (NGO2). To overcome the issue of laxness in policy enforcement, one NGO representative emphasized the critical role of the judicial courts in “stricter ruling against any offense committed by people under the influence of alcohol” (NGO2). Overall, stakeholders felt that:
“The judges should not be flexible with the laws in support of the person drinking alcohol. Those are judges that aren’t aware, that don’t have anyone to guide them so that if a drunk driver was brought to them, they shouldn’t be flexible in his favor.”.(NGO2)

Widespread corruption in the government was seen to contribute to the problem of weak governance and disregard for the rule of law:
“We sometimes issue laws, but we are lax in implementing them. What is the reason behind this laxness? I think it is because there is no transparency, lack of accountability. In my view, if we don’t have regulations that are firm and aren’t serious in enforcement, all our efforts are gone to waste”.(GO2a)

The lack of accountability was often seen to be related to differential treatment in relation to policy enforcement. One stakeholder described an example of how corruption influences enforcement:
“Even the regular policeman that stands in the road all day, cannot do his job properly. [When someone is stopped for a violation] and this person (who is stopped) calls a person in the higher echelons of government and hands the phone to the policeman [and policeman is told to dismiss the violation], this policeman is then humiliated. So, this policeman sees the people violating the law, ignores it and says to himself, why do I need to be humiliated”.(GO2a)

Weak governance and disregard for the rule of law also contributed to keeping alcohol harm-reduction policy low on the priority list of policymakers. Overall, stakeholders agreed that a sense of general disregard for this topic prevailed. A governmental organization stakeholder stated that: “decision makers, the government and the legislator must focus on the issues that the community needs, on their priorities” (GO2a), implying that alcohol control was neither a community-perceived need nor a priority for government and legislators. As a result, alcohol control policies did not receive the needed human, financial and technical resources for effective implementation. Senior GOs and NGO representatives saw parallels between alcohol control policies implementation and other public health laws, such as the tobacco control and transportation law where poor enforcement was a direct result of a lack of “sufficient number of ISF (internal security forces) trained to implement (them)” (NGO1). Limited financial resources necessary to enforce existing policies were also seen as problematic:
“The law [in reference to the transportation law that dictates penalties for drunk drivers] is present. However, the problem is the lack of resources to activate a proper mechanism of enforcement.”.(NGO1)

And another NGO stakeholder stated that the “absence of guidance for enforcement and resources [in reference to availability of alcohol breath test kits and trained security forces] to implement the law” (NGO2) was a challenge.

Stakeholders also agreed that commercial interests are often given priority over public health policymaking in Lebanon. In reference to the tobacco control policy, they noted that:
“lack of enforcement of the law [in reference to the experience of the tobacco control law] in restaurants is a result of the lobby of restaurant owners [a front for the tobacco industry] propagating the argument that enforcement will put them out of business, affect the workforce, that it is an economic burden.”.(GO2a)

A senior GO also noted the influence of alcohol-industry lobby groups, stating that:
“When the issue of tax increases was raised in the Parliament, a large group [i.e., traders and importers of alcoholic beverages] objected to this increase in taxation claiming that it adversely affects the Lebanese tourism industry.”.(GO4)

Finally, very few NGO stakeholders saw any conflict of interest issues related to interaction with or funding from alcohol industry stakeholders, an additional indicator of weak governance. While discussing possible strategies to advance an alcohol harm-reduction policy, one NGO stakeholder stated:
“Why don’t you include alcohol companies [in the discussion], because in the end, they also have strategies.”.(NGO1)

Another described his/her interactions with an alcohol industry representative as positive, “[he] will help us, he is not against us at all, he enlightened me about the topic [in reference to taxation reform]” (NGO3). An NGO reportedly sought to “work closely with a company that imports alcohol to provide ‘alcotest’ devices to be used by ISF [internal security forces] members (as a mechanism to enforce the transportation law). Unfortunately, the quantity needed was huge and they couldn’t afford buying this large quantity” (NGO1). Another described an international beer company’s road safety program as “a positive campaign” and part of non-governmental organizations “global CSR (corporate social responsibility) role” (NGO2).

### 3.3. Diverting of Responsibility towards ‘Other’ Stakeholders

Stakeholders noted that enforcement of relevant alcohol control laws was made more difficult because of the intersection of responsibility for certain legislations amongst ministries, and unclear mandates for each. The potential advertising ban was given as an example: regulating and licensing of street billboards is under the jurisdiction of the Ministry of Interior and Municipalities; but the Ministry of Information monitors TV advertisements (GO1). This patchwork of responsibility related to even one aspect of regulation creates a pattern of shifting responsibilities among ministries:
“Each ministry tells you I’m not responsible. There has to be complete coordination and complete cooperation and exchange of information and support [for the policy to be effective]”.(GO1)

More generally, however, many government stakeholders were quick to remove the blame from themselves and their institutions and place it on ‘other’ stakeholders. They described a general culture of permissiveness in Lebanon when it comes to youth alcohol drinking. This permissiveness, in their view, was propagated by parents, the educational system, and owners of alcohol serving establishments (both on-premise and off-premise). Many stakeholders saw parents as very permissive when it comes to their children’s drinking stating that:
“In the Socio-familial situation there is no control [in relation to underage alcohol drinking]”.(GO3)

With reference to the educational system, government representatives stressed the importance of schools and universities integrating this topic into a course about civic education:
“The lack of an educational curriculum in schools that raises awareness among students about the negative effects of excessive alcohol consumption, smoking and other substance use and the lack of an educational curriculum or system for teaching people about respecting regulations” was blamed for the current status quo of youth alcohol drinking.(GO4)

School-based programs that educate students about alcohol drinking were considered to be important in both private and public schools according to stakeholders. Additionally, even if educational curriculums reinforce respect for laws, “there is a need to follow up beyond this individual level because when the child grows up s/he will be surrounded by highly available and unmonitored pubs reinforcing the deviant behavior” (GO1).

Stakeholders from NGO’s and GO’s also blamed owners of alcohol serving establishments for the permissive environment that leads to unregulated youth consumption of alcoholic beverages. Different NGO representatives emphasized the responsibility of nightclubs and pubs/bars to control underage drinking as well as excessive drinking and noted the importance of imposing consequences for violations—particularly in relation to underage drinking. As one NGO interviewee put it:
“It is the responsibility of nightclub owners [to prevent underage youth to come in], it is their responsibility, they also should be aware, they shouldn’t say that they aren’t responsible and to get the security forces; No in the end [they] are responsible”.(NGO1)

### 3.4. Views of Alcohol Industry Stakeholders

We analyzed results from interviews with alcohol industry stakeholders separately as they have unique interests in alcohol control policy making that we argue trump all other interests. Table 2 summarizes the strategies noted by industry representatives that we interviewed.

With regard to regulating advertising and promotion, industry stakeholders had several arguments against it, stating that this form of regulation (i) is not effectiveness; and (ii) could harm the economy. They further noted that—in their opinion, advertising and promotion does not encourage irresponsible drinking. One industry stakeholder noted:
“Before we start applying it [in reference to banning advertising of alcohol products] in Lebanon, let us see studies that prove to the people that when you do this, you have positive results.”.(AI6)

Furthering his/her argument, he/she continued:
“I am not with banning advertisements because you can’t tell them not to advertise on the roads. And same goes for the TVs and if they don’t see there, then they will see it on the internet. I don’t think the advertisements are the main problem.”.(AI6)

And in reference to banning advertisements of wine products specifically, he/she argued against it because “The advertising sector in Lebanon is on the verge of collapse, banning wine commercials is detrimental to the economy” (AI6). Propagating the industry’s argument related to the victim-blaming concept of ‘responsible drinking’ one stakeholder stated:
“In addition, we are against banning advertisement because if alcohol is overused then it is harmful, but if it is used wisely then it has many health benefits.”.(AI4)

Overall, consensus was strong against banning of advertisements. In fact, as an alternative to the suggested banning of advertising and promotion, industry stakeholders suggested equal time for counter ads as an alternative where, “in return [companies] should give equal funding to the Ministry of Public Health (MOPH) to do a counter ad to spread awareness on the dangers of alcohol” (AI2). Another interviewee suggested a similar approach: ”5% of the profit from advertisement should be taken by the government to do good things such as research, and civic engagement. (A14).

Stakeholders also discussed labelling policies, and were quick to discuss the value they see in self-regulation. One noted, that they had “added in Arabic: ‘Don’t drink and drive’, and ‘not allowed under 18” (AI3) as labels on energy vodka mixes. The government had reacted to energy vodka drink by cracking down on them. This same stakeholder stated: “we cooperated with Ministry of Economics, and Ministry of Public Health, and we reached this result”, in reference to labelling alcohol energy drinks (AI3). Alcohol industry stakeholders stated that warning such as these were necessary:
“Every alcohol advertisement must include a warning on not over drinking and a warning about pregnant women.”.(A14)

However, not all industry stakeholders agreed with the importance of warning labels, justifying lack of warning labels by using the responsible drinking argument. As one interviewee put it:
“In my opinion, you as people of science, are you able to put a label on it like you put it on smoking? Like ‘be careful smoking is deadly’, you can’t put on a bottle of whisky that ‘don’t drink whisky’, that the whisky will cause diseases, you can’t put it. Too much [drinking] makes one lose balance, lose oneself, lose one’s personality, yes, but you can’t say [on alcohol bottles] what you say [on cigarette packs] about smoking.”.(AI1)

Related to serving alcohol to minors, industry stakeholders shifted the blame for disrespecting the laws to bar owners, barmen, bouncers, and waiters. According to one stakeholder:
“You can’t be sure if the customers are underage especially because nobody asks them about their ID at the entrance and this is compounded by the availability of fake IDs with minors.”.(AI5)

The ministry of tourism and the touristic police were blamed for not enforcing the law. Another alcohol industry stakeholder raised concern over enforcement of an underage drinking law stating that it might result in venues losing business. As highlighted in the themes above, alcohol industry stakeholders also viewed enforcement of the law by the judicial system as problematic, since the fines for violations were negligible which often reinforced lack of compliance. Industry representatives were in favor of increasing fines against those who disrespect the law:
“We need fines to be stricter. The current fines are so very minimal that you can bribe an ISF [internal security force] member to avoid the hassle of paying a penalty.”.(AI6)

Industry stakeholders were the only ones that elaborated on concerns with taxation policy. The representative of the local producers saw the existing weak taxation structure on imported alcoholic products as “posing a direct threat to local production and causing the closure of many local factories and a decrease in produced quantities” (AI4). In reference to taxing local beer products one industry interviewee said that:
“You must sell a bottle of beer at low prices because it is a local product, we are proud of it and we want to sell it to everyone, everyone has the right to drink beer”.(AI)

Local alcohol producers warned about issuing any arbitrary laws regulating alcohol taxation because “when there is prohibition then there is the creation of a black market” (AI4). Other industry stakeholders echoed this negative view of taxation. Taxation was not seen as a positive proposition at all nor a way that will limit consumption. It was also viewed as an unfavorable policy option:
“Because you are affecting the night clubs in Lebanon today you are affecting the customer, instead of going to enjoy a drink, we are a touristic destination, a musical destination, it is very important in the Arab world, and the whole world. Beirut is a night life destination it is not in our interest to do this, let us organize the sector and close all these small shops that sell the alcohol.”.(AI5)

Another alcohol industry interviewee, in reference to a proposed change in taxation structure, referred to an incident where they successfully lobbied against a proposed change:
“So, we moved forward and thank God we talked to the parliament members in charge (of the proposed change in taxation regulation) and we were able to reach where we should reach. We are like someone who has a family, but the father does not care for them. We didn’t have a caretaker for this family that is in the parliament.”(AI3)

The industry saw an increase in taxation as problematic, citing that:
“Every time you increase taxes, you open the door for smuggling, so we can’t hide behind our finger, the borders are open everywhere, today we are hearing on TV that they’re saying if customs are high then the doors to smuggling will be open, now, they raised the prices 5 times so tomorrow people will be born to work in smuggling and this will harm us more later because this kind of corruption refreshes the illegitimate competition and harms the people that want to work with ethics, and according to regulations and by the law. So, this opening way to corruption creates illegal competition that harms the sector so if the government isn’t able to contain its borders 100%, then it is better not to raise the customs.”.(AI2)

The interference of the alcohol industry and their allies in policy making that is described above was also confirmed by the other interviewed stakeholders as noted in the above themes.

Alcohol industry interviewees in fact described collaborations with civil society to promote responsible drinking as a strategy they follow:
“We are interested today that youth that drink, don’t just drink and stop, but drink and keep it up. This is why we are interested that youth drink moderately, know when to stop. We contacted NGO’s who are interested in youth and we are going to soon have an agreement and collaboration with them”.(AI3)

This collaboration and lack of regard to conflict of interest was also noted above through interviews with the other stakeholders. Finally, as with other stakeholders, alcohol industry stakeholders also considered the problem of youth alcohol drinking to be mainly a family disciplinary matter rather than a policy issue of their responsibility emphasizing that:
“Orientation should start from home first and parents have a responsibility towards knowing about their children’s outings, the places they go to, how long they are allowed to stay outside home and what they actually consume.”.(AI1)
“Some families encourage their children to drink a glass” noting the need “to start raising awareness at this point start at home.”.(AI2)

## 4. Discussion

This research highlighted the views of key stakeholders from government and civil society organizations regarding a national alcohol harm-reduction policy to reduce harmful youth drinking in Lebanon. The themes that emerged from the interviews were centered around three points. First, that the state of alcohol control policies in place was inadequate; second, that weak governance and disregard for the rule of law contributed to that state; and third, diverting of the blame for the problem to ‘others’. Alcohol industry stakeholders shed doubt on effectiveness, relevance, or feasibility of various alcohol harm-reduction policies and proposed alternative approaches including self- regulation. Industry stakeholders also cited the weak governance framework; and diverted responsibility to parents, family and other stakeholders.

Other research has corroborated the highlighted inadequacy of alcohol harm-reduction policies in Lebanon. A review of alcohol policies in Lebanon as compared to WHO core policies revealed the extent of laxness of these policies. This was further confirmed in focus group discussions with youth who called for increased and improved government involvement in policy making for alcohol control [43]. The inadequacy of the legal framework for alcohol is not unique. Other health-related policies in Lebanon are similarly either weakly formulated, or weak in implementation. Regarding the latter, one clear example also highlighted by the interviewees is the tobacco control law [44]. Wavering government commitment to alcohol regulation policy has also been documented in other countries. Varvasovsky et al. (1998) interviewed stakeholders in Hungary and reported the lack of an alcohol harm-reduction policy, and the absence of intersectorality in dealing with this issue. This echoes our findings that part of the problem is the distribution of various aspects of the alcohol control legal framework between various ministries. In the UK for example, government entities were competing amongst themselves to regulate alcohol and managed to sabotage the alcohol control policy process in order to protect their own interests [45].

Both government and alcohol industry stakeholders emphasized the responsibility of parents in monitoring their children’s behavior in relation to alcohol use; and the role of schools in raising awareness of the harms of alcohol drinking among youth. Lebanon’s school health curriculum does include instruction on the harms of alcohol. However, this content begins in 9th grade, despite clear indications that alcohol use is starting at a much younger age [3]. The extent to which the topic is actually covered by teachers is also unknown. Similar findings of shifting of blame were reported in a study of stakeholders in South Africa [46]. This shift of blame to the parents and the educational establishment carries significant meaning in policy rationale; it shifts focus from the “structural causes” related to availability, accessibility and advertisement to “individual responsibility” for drinking [32]. It also promotes the weakest form of intervention based on awareness and education, a non-evidence based approach to alcohol harm reduction policy [47]. These interventions have proven to be ineffective in controlling harm and are largely in tune with industry strategies [47,48].

The industry stakeholder positions noted in Table 2 mimic those used by the industry elsewhere [47,49,50,51,52]. In addition, the findings pointed to NGOs’ failure to recognize potential conflicts of interest between public health policies and the alcohol industry. Industry funding of local NGOs anti-drinking and driving national campaigns [3], as well as sports and youth events is common in Lebanon. Globally, alcohol industry stakeholders have funded governmental and non-governmental alcohol awareness or policy campaigns to further their interests. For example, an analysis of “stop out-of-control drinking campaign” which is funded by Diageo in Ireland concluded that the campaign frames alcohol drinking as a behavioral issue and supports solutions that are not in line with public health evidence [53].

Our findings should be interpreted in light of a few limitations. This manuscript reports on the positions and views of stakeholders who accepted our invitation to be interviewed, which might reflect an inherent bias, possibly related to our identity as academics from a school of public health. For example, owners of one of the leading advertising companies for the alcohol sector refused to meet with us, claiming that such a meeting would not be convenient for them. We did reach stakeholders in each of the WHO identified core intervention areas: availability, affordability, advertising and marketing, and drink driving. They provided deep insight into the current situation of alcohol control harm reduction in Lebanon, and as noted in the results, also provided insight into perceptions of other stakeholders. Therefore, overall, we do feel that the results are broadly representative of the larger list of stakeholders. Also, though we report alcohol industry stakeholder views, we did not reach the comprehensive definition of this group as a “multinational business complex that includes not only the producers of beer, wine, and distilled spirits but also a large network of distributors, wholesalers, and related industries, such as hotels, restaurants, bars, and advertisers” [54]. Though parents were often brought up as a key source of the alcohol use problem of youth, we neither interviewed the parents nor the youth or educational system stakeholders. In addition, all views expressed by our stakeholders are their own perceptions of the situation, informed by their expertise and position, and may or may not be ‘accurate’. The objective of a stakeholder analysis is precisely to understand the views/positions of stakeholders vis-à-vis a particular policy in order to understand the likelihood that the policy in question will be passed, and/or the challenges that must be overcome to create an enabling environment for the passage of the policy. The perceptions of participants—whether accurate or not—provide feedback to researchers, advocates, and decision makers regarding strategies that must be implemented to create the enabling environment for the passage of a law.

## 5. Conclusions

Our findings indicate that alcohol harm-reduction policies are far from becoming a policy priority in Lebanon. There is a clear need to shift the narrative from victim blaming to structural conditions. The interviewed stakeholders provided some broad ideas for ways forward to create a more enabling environment for the passage of an alcohol control harm reduction policy for youth, but more specific attention to evidence-based strategies to counteract negative positions is needed. Community and media mobilization may be an important tool to achieve this goal, and to begin to stifle and counteract the industry’s individual responsibility narrative. Globally, advocacy groups such as Mothers Against Drunk Driving (MADD) have been successful in putting and keeping this topic on the agenda thus influencing the policy agenda. Partnerships between academics who can provide the evidence-base and NGO and media outlets who can engage in social action can be successful. More importantly, strategies to counteract weak governance and disregard for the rule of law will be critical if progress in developing and enforcing an inter-sectoral alcohol control evidence-based policy will ever come to fruition in Lebanon. Engaging and training judges may be one way to shift the tide in this direction [55]. Intersectoral coalition building among governmental ministries with a focus on win-win policies will also be important [56]. As a strategy to begin to shift the narrative, the research team formed a coalition of individuals (some of whom were interviewed as stakeholders) committed to passage of a stronger alcohol control policy in Lebanon. The coalition met three times over the course of two years to discuss local evidence of alcohol use among youth, as well as stakeholder positions, and evidence of policies that are critical for harm reduction. The coalition encouraged the research team to develop a policy brief, and engage in a policy dialogue around evidence-based policies for youth alcohol use harm reduction [57]. This dialogue took place in January 2017, and ended with agreement by all present on next steps including continued advocacy by the coalition. Researchers were also urged to continue to develop evidence to support the passage of a law [58]. This is the first study reporting stakeholders’ views of alcohol control harm-reduction policies in Lebanon and in the Arab world. Future research should test possible interventions to influence stakeholders’ views regarding alcohol harm-reduction policy. With data showing increasing use of alcohol by young people, at a younger age, and closing of the gap in gender use, advancing evidence-based alcohol control harm reduction policy is imperative [24,27].

## Figures and Tables

**Table 1 ijerph-16-02874-t001:** List of organizations and institutions from which stakeholders participated in the interviews.

Stakeholder	Quote Label
1. The Ministry of Industry	GO* 1
2. The Ministry of Social Affairs	GO2a
3. The Ministry of Social Affairs	GO2b
4. The Ministry of Tourism	GO 3
5. The Ministry of Youth and Sports	GO 4
6. The Ministry of Economy and Trade (Customer Protection Office)	GO5
7. The Ministry of Information	GO 6
8. The Ministry of Justice	GO 7
9. The Ministry of Finance	GO 8
10. The Ministry of Public Health	GO 9
11. Kunhadi	NGO * 1
12. YASA	NGO 2
13. JAD (Jeunesse-Anti-Drogue)	NGO 3
14. Mentor Arabia	NGO 4
15. Skoun- Lebanese Addictions Center	NGO 5
16. MENAHRA	NGO 6
17. Syndicate of Importers of Foodstuff, Consumer Products & Drinks in Lebanon	AI * 1
18. Syndicate of Lebanese Food Industrialists	AI 2
19. Syndicate of Alcohol Importers in Lebanon	AI 3
20. Syndicate of Alcoholic Beverages Industries in Lebanon	AI 4
21. Syndicate of Owners of Restaurants, Cafes, Night-Clubs and Pastries in Lebanon	AI 5
22. Union Vinicole du Liban: (UVL)	AI 6

* GO = governmental organization; NGO = non-governmental organization/Civil society, AI = alcohol industry.

**Table 2 ijerph-16-02874-t002:** Views and related quotes of the alcohol industry stakeholders.

Alcohol Industry Stakeholder Views	Supporting Quotations
**Advertising Bans Are Not Effective**	Before we start applying it [in reference to banning advertising of alcohol products] in Lebanon, let us see studies that prove to the people that when you do this, you have positive results. I am not with banning advertisements because you can’t tell them not to advertise on the roads. And same goes for the TVs and if they don’t see there, then they will see it on the internet. I don’t think the advertisements are the main problem.” (Alcohol Industry –AI -6). “The advertising sector in Lebanon is on the verge of collapse, banning wine commercials is detrimental to the economy” (AI6).
**Promoting Responsible Drinking**	“In addition, we are against banning advertisement because if alcohol is overused then it is harmful, but if it is used wisely then it had many health benefits “(AI4.). “…Too much [drinking] makes one lose balance, lose oneself, lose one’s personality, yes, but you can’t say [on alcohol bottles] what you say [on cigarette packs] about smoking.” (AI1). “We are interested today that youth that drink, don’t just drink and stop, but drink and keep it up. This is why we are interested that youth drink moderately, know when to stop. We contacted NGO’s who are interested in youth and we are going to soon have an agreement and collaboration with them” (AI3).
**Self-Regulation**	We “added in Arabic: ‘Don’t drink and drive’, and ‘not allowed under 18′ …we cooperated with Ministry of Economics, and Ministry of Public Health, and we reached this result”.
**Importance of Alcohol Business to the Economy**	“Because you are affecting the night clubs in Lebanon today, you are affecting the customer, instead of going to enjoy a drink, we are a touristic destination, a musical destination, it is very important in the Arab world, and the whole world. Beirut is a night life destination it is not in our interest to do this, let us organize the sector and close all these small shops that sell the alcohol”. (AI5)
**Others’ Behavior is the Problem**	“You can’t be sure if the customers are underage especially because nobody asks them about their ID at the entrance and this is compounded by the availability of fake IDs with minors.” (AI5). “Orientation should start from home first and parents have a responsibility towards knowing about their children’s outings, the places they go to, how long they are allowed to stay outside home and what they actually consume” (AI1). “Some families encourage their children to drink a glass” noting the need “to start raising awareness at this point … start at home.” (AI2). “We need fines to be stricter. The current fines are so very minimal that you can bribe an ISF [internal security force] member to avoid the hassle of paying a penalty.” (AI6).
**Labelling of Alcohol Products Is Not Effective**	“Every alcohol advertisement must include a warning on not over drinking and a warning about pregnant women.” (A14). “In my opinion, you as people of science, are you able to put a label on it like you put it on smoking? Like ‘be careful smoking is deadly’, you can’t put on a bottle of whisky that ‘don’t drink whisky’, that the whisky will cause diseases, you can’t put it. Too much [drinking] makes people lose balance, lose himself, lose his personality, yes, but you can’t say [on alcohol bottles] what you say [on cigarette packs] about smoking” (AI1).
**Taxation of Alcohol Products Is Not Effective**	The existing weak taxation structure on imported alcoholic products was seen as “posing a direct threat to local production and causing the closure of many local factories and a decrease in produced quantities” (AI4). “You must sell a bottle of beer at low prices because it is a local product, we are proud of it and we want to sell it to everyone, everyone has the right to drink beer” (AI). *“when there is prohibition then there is the creation of a black market”* (AI4). “Every time you increase taxes, you open the door for smuggling, so we can’t hide behind our finger, the borders are open everywhere, today we are hearing on TV that they’re saying if customs are high then the doors to smuggling will be open, now, they raised the prices 5 times so tomorrow people will be born to work in smuggling and this will harm us more later because this kind of corruption refreshes the illegitimate competition and harms the people that want to work with ethics, and according to regulations and by the law. So, this opening way to corruption creates illegal competition that harms the sector so if the government isn’t able to contain its borders 100%, then it is better not to raise the customs.” (AI2).
**Corporate Social Responsibility (CSR) Programs**	”5% of the profit from advertisement should be taken by the government to do good things such as research, and civic engagement. This is not an arbitrary policy, but a consensus” (A14).

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
