# Peer review of "“Everyone Has the Right to Drink Beer”: A Stakeholder Analysis of Challenges to Youth Alcohol Harm-Reduction Policies in Lebanon"

_ijerph, 2019, doi:10.3390/ijerph16162874_

Round 1
Reviewer 1 Report
This is a clear and straightforward study, filling an obvious gap in the literature.
It does need some copy-editing to make the English more colloquial. I noted, for instance, that the sentences at lines 324, 329 and 348-9 needed minor change. "CSR" in the line above 284 should be spelled out; Corporate Social Responsibility. Two quotes in Table 2 are repeated in the main text: the 4th quote at lines 341-344. the quote 3rd from the bottom at lines 315-318. Reference 25: Organization, W.H. --> World Health Organization.
The authors might consider a paragraph at the beginning giving some context concerning Lebanon and alcohol: that it is a multicultural country marked by strong splits on religious affiliation, with the religions having very different teachings on alcohol. That is was a French colony until (date), and presumably inherited French-influenced laws and regulatory traditions on alcohol; that it has had a troubled history in recent decades of communal and international disputes. The authors can write this much better than I can; it provides some context to the complaints of the informants about the lack of regulations and ineffectiveness in the enforcement of those that do exist. It might be worth considering a paragraph also at the conclusion on potential paths forward, including anything informants may have had to say about this..
Reviewer 2 Report
This is a good study and appropriate for this journal. It is original and novel. However, to strengthen the article, I have several suggestions and comments:
(1) As stated in the limitations (page 11, lines 415-423), these are just opinions of a select few government, non-government and alcohol industry representatives who agreed to participate. They are not representative of all stakeholders. While some useful information came out of the interviews, some of the opinions should be verified.
(2) The reader needs a quantification of the problem in Lebanon. What is the alcohol consumption rate per capita in Lebanon? How does that compare to Israel? To other Middle Eastern countries? To other countries around the world?
(3) What is the minimum legal drinking age in Lebanon? 18? What proportion of youth under age 18 report drinking alcohol in the past 30 days or past year? Report binge drinking? Report drinking before age 14?
(4) What proportion of Lebanon's population are addicted to alcohol? What proportion of traffic fatalities in Lebanon involve alcohol-impaired drivers? There must be some quantitative information that provides a picture of alcohol in Lebanon. Please report it in the Introduction.
(5) What does Lebanon's minimum legal drinking age law say? Illegal to purchase? Possess? Consume? Use a fake ID to purchase? To sell to an underage? There are 20 components to the drinking age law in the United States.
(6) You must strengthen this manuscript by providing basic data on the alcohol problem in Lebanon, otherwise the reader has nothing to compare your findings with.
